# LV-XATTN: Distributed Cross-Attention for Long Visual Inputs in Multimodal Large Language Models

**Tzu-Tao Chang** [1]   **Shivaram Venkataraman** [1]

## Abstract

Cross-attention is commonly adopted in multimodal large language models (MLLMs) for integrating visual information into the language backbone. However, in applications with large visual inputs, such as video understanding, processing a large number of visual tokens in cross-attention layers leads to high memory demands and often necessitates distributed computation across multiple GPUs. Existing distributed attention mechanisms face significant communication overheads, making cross-attention layers a critical bottleneck for efficient training and inference of MLLMs. To address this, we propose LV-XAttn, a distributed, exact cross-attention mechanism with minimal communication overhead. We observe that in applications involving large visual inputs, the size of the query block is typically much smaller than that of the key-value blocks. Thus, in LV-XAttn we keep the large key-value blocks locally on each GPU and exchange smaller query blocks across GPUs. We also introduce an efficient activation recomputation technique to support longer visual context. We theoretically analyze the communication benefits of LV-XAttn and show that it can achieve speedups for a wide range of models. Our evaluations with Llama 3-V, mPLUG-Owl3 and OpenFlamingo models find that LV-XAttn achieves up to $10.62\times$ end-to-end speedup compared to existing approaches.

## 1. Introduction

Large language models (LLMs) have shown exceptional performance in language processing tasks that involves long context, such as long document understanding (Sun et al., 2024; Bertsch et al., 2023) and repository-level code completion (Shrivastava et al., 2023; Zhang et al., 2023). Their strong reasoning capabilities have motivated efforts to expand beyond language inputs, giving rise to multimodal large language models (MLLMs). These models can process and reason about other modalities, such as visual inputs, enabling applications like video understanding (Qian et al., 2024; Islam et al., 2024; He et al., 2024) and image processing (Guo et al., 2024; Li et al., 2023b).

A common approach to integrating visual inputs into LLMs is through *cross-attention* (Alayrac et al., 2022; Laurençon et al., 2024; Li et al., 2023a; Ye et al., 2024; Grattafiori et al., 2024; Dai et al., 2024), where queries derived from the text input interact with keys and values derived from the visual inputs. This enables effective fusion of multimodal information. In MLLMs, cross-attention layers are inserted between language model blocks, enabling the LLM to process intermediate representations that are integrated with visual information.

However, the memory requirement of cross-attention layers is a limiting factor for applications involving large visual inputs, such as long video understanding. For example, in Llama 3-V (Grattafiori et al., 2024), applying cross-attention to a text sequence of length 2048 and a 20-minute video sampled at 1 frame per second (fps) requires over 234 GB of memory, even with a memory-efficient attention implementation (Dao, 2024). This exceeds the memory capacity of existing accelerators, necessitating the distributed computation of the attention operation across multiple workers.

Existing distributed attention approaches can be categorized into two classes: head-parallelism and sequence-parallelism. Head-parallelism methods such as Deepspeed-Ulysses (Jacobs et al., 2024) and Megatron-LM (Korthikanti et al., 2023a) partition the computation along the head dimension of multi-head attention. Consequently, maximum degree of parallelism is capped by the number of heads used in multi-head attention. This translates to an upper bound in terms of memory capacity, preventing them from processing longer visual inputs which have memory demands beyond this. In addition, the number of workers must be divisible by the total number of heads to ensure a balanced load across workers. Otherwise, resource underutilization may occur due to stragglers. On the other hand, sequence parallel methods

[1]University of Wisconsin-Madison. Correspondence to: Tzu-Tao Chang <tchang85@wisc.edu>.

*Proceedings of the 42nd International Conference on Machine Learning*, Vancouver, Canada. PMLR 267, 2025. Copyright 2025 by the author(s).

such as Ring Attention (Liu et al., 2024a) partition the computation along the input sequence dimension, overcoming the limitation of head-parallelism methods. However, when applied to cross-attention with large visual inputs, these approaches suffer from large communication overheads even after overlapping computation and communication. Figure 2 shows that cross-attention operations distributed with Ring Attention can account for up to 88% of the iteration time, despite comprising only 3% of the total parameters.

In this work, we present LV-XAttn, a distributed, exact cross-attention mechanism that employs sequence-parallelism *with minimal communication overhead*. Our main observation is that while keys and values derived from visual inputs are large, the queries derived from text input are typically small in MLLMs. For example, in the video understanding benchmark Video-MME (Fu et al., 2024), an input processed with Llama 3-V results in an average sequence length of 15,279,944 for keys and values and 5,514 for queries, when frames are sampled at 1 fps. Based on this, LV-XAttn organizes each worker to locally store a partition of the large key and value blocks, while small query blocks are transmitted between workers to compute the attention output in a blockwise fashion. This significantly reduces the communication volume compared to Ring Attention. For instance, on the Video-MME benchmark, LV-XAttn reduces communication volume to just 0.04% of that required by Ring Attention. Furthermore, the reduced communication can be effectively overlapped by computation, allowing distributed cross-attention to be performed without incurring any communication overhead.

To further enable the processing of longer visual inputs, we employ an activation recomputation technique that is specific to MLLMs. In standard attention implementations, activations including queries, keys, and values need to be saved for backward pass (Korthikanti et al., 2023b). Storing the large key and value tensors for every cross-attention layer introduces additional memory pressure. We observe that since cross-attention layers in MLLMs share the same input visual tokens, we can maintain a single copy of the visual tokens accessible to all cross-attention layers and recompute activations during the backward pass. This allows us to process up to $1.6\times$ longer visual inputs with just less than 8% overhead in terms of iteration time.

We perform comprehensive evaluation of LV-XAttn on Llama 3-V, mPLUG-Owl3 (Ye et al., 2024) models and OpenFlamingo (Awadalla et al., 2023) models across multiple cluster configurations, including setups with A100 and A30 GPUs. LV-XAttn speeds up the cross-attention operation by up to $45.85\times$ and overall model iteration time by up to $10.62\times$ compared to Ring Attention. By minimizing communication volume and further overlapping communication with computation, we demonstrate that LV-XAttn

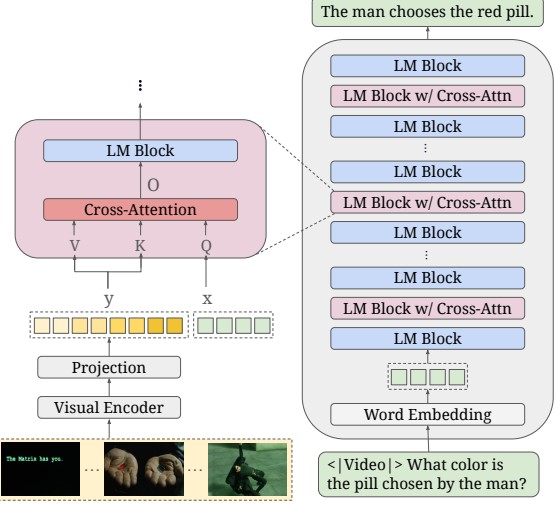

*Figure 1.* MLLM with cross-attention.

incurs less than 0.42% overhead compared to the theoretical no-communication baseline. LV-XAttn is available at `https://github.com/uw-mad-dash/LV-XAttn`.

## 2. Background

**Cross-attention** Cross-attention (Vaswani et al., 2017) is a variant of self-attention to model interactions between different sequences. The input to cross-attention consists of two sequences $x \in \mathbb{R}^{S_Q \times d_{\text{embed}}}$ and $y \in \mathbb{R}^{S_{KV} \times d_{\text{embed}}}$, where $S_Q$ and $S_{KV}$ denote the sequence lengths of $x$ and $y$, respectively, and $d_{\text{embed}}$ is the embedding dimension. The input sequence $x$ is multiplied with the projection matrices $W_Q \in \mathbb{R}^{d_{\text{embed}} \times d}$ to obtain the queries $Q \in \mathbb{R}^{S_Q \times d}$, while the input sequence $y$ is multiplied with the projection matrices $W_K, W_V \in \mathbb{R}^{d_{\text{embed}} \times d}$ to obtain the keys and values $K, V \in \mathbb{R}^{S_{KV} \times d}$, where $d$ is the hidden dimension. The attention output $O \in \mathbb{R}^{S_Q \times d}$ is then computed as:

$$O = \text{softmax}(\frac{QK^T}{\sqrt{d}})V$$

**Multimodal Large Language Models** As LLMs continue to evolve, researchers are investigating how to incorporate vision and other modalities into these models. One common way is to embed cross-attention layers into the language model. This design has been adopted by a number of models including Flamingo (Alayrac et al., 2022), Otter (Li et al., 2023a), mPLUG-Owl3 (Ye et al., 2024), IDEFICS (Laurençon et al., 2024), Llama 3-V (Grattafiori et al., 2024), NVLM-X (Dai et al., 2024), and NVLM-H (Dai et al., 2024). Broadly, these models follow the architecture illustrated in Figure 1. They include a visual encoder, a projection layer, and an LLM. Cross-attention layers are

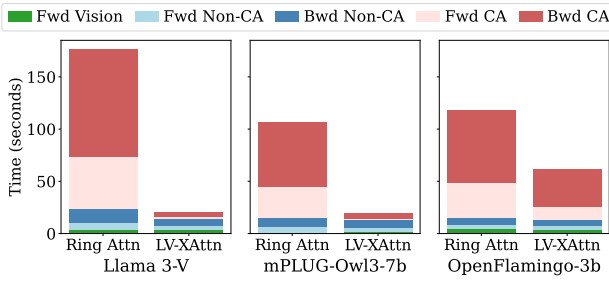

*Figure 2.* Runtime breakdown for a single iteration of Llama 3-V, mPLUG-Owl3-7b, and OpenFlamingo-3b using Ring Attention and LV-XAttn on 16 A100 GPUs. LV-XAttn reduces the time spent on cross-attention computation by 96%, 93%, and 53% for the three models, respectively, compared to Ring Attention. "FWD Vision" refers to the forward pass through the vision encoder and the projection layer; "FWD CA" and "BWD CA" refer to the forward and backward passes through the cross-attention layers in the LLM; and "FWD Non-CA" and "BWD Non-CA" refer to the forward and backward passes through the non-cross-attention layers in the LLM. Llama 3-V was evaluated with a text length of 1K and a frame count of 192 ($S_Q = 1$K, $S_{KV} = 1200$K); mPLUG-Owl3-7b was evaluated with a text length of 4K and a frame count of 2K ($S_Q = 4$K, $S_{KV} = 1458$K); and OpenFlamingo-3b was evaluated with a text length and a frame count of 32K ($S_Q = 32$K, $S_{KV} = 2048$K).

interleaved between the layers of the LLM. Language inputs are fed directly into the LLM and the resulting intermediate representations are passed to the cross-attention layers as $x$. Visual inputs are processed by the visual encoder and the projection layer to produce visual tokens, which are then passed to the cross-attention layers as $y$, enabling the incorporation of visual information.

**Challenges with Large Visual Inputs** When applied to scenarios with large visual inputs, cross-attention requires significant memory resources and therefore presents a scaling challenge. The standard implementation of attention involves materializing the matrix product $QK^T \in \mathbb{R}^{S_Q \times S_{KV}}$, resulting in memory complexity that scales with the product of text sequence and the visual sequence length. The large amount of visual tokens from long videos inputs thus causes a memory bottleneck. For example, in Llama 3-V, a 20-minute video sampled at 1 fps is encoded to a visual input with more than 7 million tokens.

While memory-efficient methods like FlashAttention (Dao et al., 2022; Dao, 2024) reduce the memory footprint of attention operations to enable handling longer context lengths, the amount of memory required still often surpasses the capacity of a single worker. For example, processing a 20-minute long video with a language input of 2048 tokens in Llama 3-V demands over 234 GB of memory for the cross-attention operation, even with FlashAttention.

To handle large visual inputs, distributed attention approaches have been proposed. These methods can be categorized into two classes: head-parallelism and sequence-parallelism. Head-parallelism methods such as Deepspeed-Ulysses (Jacobs et al., 2024) and Megatron-LM (Korthikanti et al., 2023a) distribute the computation of different attention heads across multiple workers. However, their scalability is limited by the number of attention heads, which imposes an upper bound on memory capacity and thus maximum sequence length.

To overcome this limitation, sequence-parallelism methods such as Ring Attention (Liu et al., 2024a) propose distributing the attention operation across multiple workers along the sequence dimension. Specifically, with $n$ workers, each worker $i$ is responsible for storing one block of query, key and value $Q_i \in \mathbb{R}^{\frac{S_Q}{n} \times d}, K_i \in \mathbb{R}^{\frac{S_{KV}}{n} \times d}, V_i \in \mathbb{R}^{\frac{S_{KV}}{n} \times d}$ and computing the attention block $O_i \in \mathbb{R}^{\frac{S_Q}{n} \times d}$. Computation of $O_i$ can be decomposed to

$$O_i = \text{softmax}(\frac{Q_i[K_0, ..., K_{n-1}]^T}{\sqrt{d}})[V_0, .., V_{n-1}]$$

$O_i$ is computed iteratively by transmitting key-value blocks among workers in a ring-like fashion. To facilitate the block-wise computation of $O_i$, worker $i$ has to maintain necessary softmax statistics $L_i \in \mathbb{R}^{\frac{S_Q}{n}}$. During round $r$, worker $i$ computes partial attention using the blocks $Q_i$, $K_{(i-r) \mod n}$, and $V_{(i-r) \mod n}$ and updates $O_i$ and $L_i$. The key-value block is then sent to worker $i + 1$ while a new key-value block is received from worker $i - 1$.

While Ring Attention can be used to distribute the cross-attention operation, the presence of large key-value blocks makes Ring Attention communication-bound, resulting in highly inefficient cross-attention operations. As illustrated in Figure 2, despite comprising only 3% of the total parameters, cross-attention operations can account for up to 88% of the iteration time when using Ring Attention.

## 3. LV-XAttn: Distributed Cross-Attention with Minimal Communication Overhead

In this section, we introduce LV-XAttn, our method for efficiently distributing the cross-attention operation with minimal communication overhead. We also present an activation recomputation technique specific to MLLMs to reduce memory pressure, enabling the processing of longer visual contexts.

### 3.1. Method

The primary observation that motivates our work is that, in applications involving large visual inputs, the size of the *query block is typically much smaller* than that of the key-

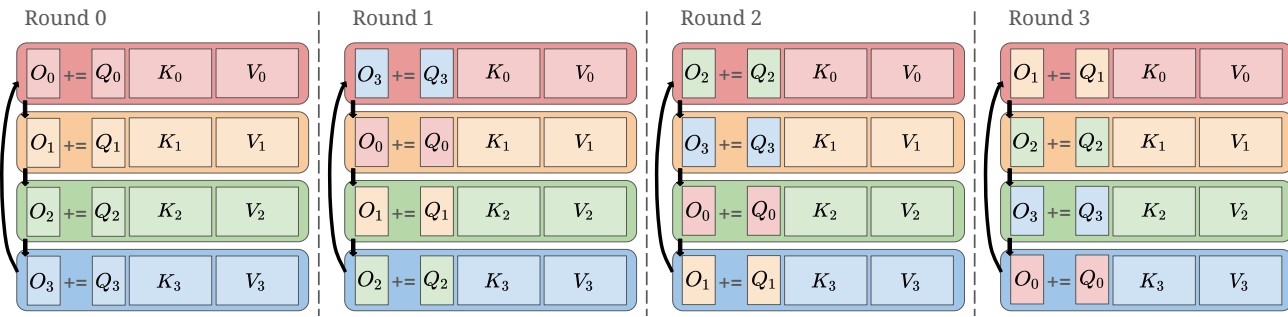

*Figure 3.* LV-XAttn with 4 workers. We partition the KV blocks and each worker stores their respective large key-value blocks $K_i, V_i$. We also partition the query ($Q_i$), output ($O_i$), and softmax statistics ($m_i$ and $l_i$ omitted in the figure). The query and output are rotated among workers to compute the attention.

---

**Algorithm 1** LV-XAttn Forward Pass for Worker $i$

---

  **Input:** data $Q_i, K_i, V_i$
  Initialize $O_i, L_i \leftarrow 0$
  **for** $round = 0$ **to** $n - 1$ **do**
    $j_{\text{prev}} \leftarrow (i - round + 1) \bmod n$
    $j \leftarrow (i - round) \bmod n$
    $j_{\text{next}} \leftarrow (i - round - 1) \bmod n$
    **do in parallel:**
      Send $O_{j_{\text{prev}}}, L_{j_{\text{prev}}}, Q_j$ to worker $(i + 1) \bmod n$
      Recv $O_j, L_j, Q_{j_{\text{next}}}$ from worker $(i - 1) \bmod n$
      $\Delta O, \Delta L \leftarrow$ FlashAttention$(Q_j, K_i, V_i)$
    $O_j, L_j \leftarrow$ Rescale$(O_j, L_j, \Delta O, \Delta L)$
  **end for**

---

value blocks. For instance, in the widely-used video understanding benchmark Video-MME (Fu et al., 2024), videos have an average duration of 2,386 seconds. Each frame is encoded by the visual encoder and projection layer in an MLLM into multiple visual tokens. For example, Llama 3-V generates 6404 visual tokens per frame. With a sampling rate of 1 fps, each video results in a key-value sequence length of $S_{KV} = 15279944$. On the other hand, the average text prompt for long videos consists of 3128 words, including the question, options, answer, and subtitles, resulting in a query sequence length of $S_Q = 2386 + 3128 = 5514$. As a result, distributed attention mechanisms that involves movement of key-value blocks incur substantial communication overhead.

To address this, we propose LV-XAttn, which keeps the large key-value blocks locally on each worker, while smaller query blocks, attention blocks, and necessary softmax statistics are exchanged among workers in a ring-style fashion. This is illustrated in Figure 3. During each round, each worker $i$ computes attention using its local key-value blocks $K_i$ and $V_i$ and query blocks $Q_j$ received from peers. This computation generates partial attention blocks $\Delta O$ and partial softmax statistics $\Delta L$. The worker then updates the

received attention block $O_j$ and softmax statistics $L_j$ by rescaling them using $\Delta O$ and $\Delta L$. The worker then sends $Q_j, O_j$ and $L_j$ to the next worker in the ring topology and receives $Q_{j-1}, O_{j-1}$ and $L_{j-1}$ from the previous worker. After $n$ rounds, the computed attention block $O_i$ and softmax statistics $L_i$ are returned to worker $i$.

**Overlapping Computation and Communication** To further reduce communication overhead, we can overlap the attention computation with data transmission between workers. While performing attention computation with $Q_j, K_i$ and $V_i$, worker $i$ also does the following in parallel

- Receive $O_j$ and $L_j$ from worker $i-1$, which are needed for rescaling in this round.

- Receive $Q_{j-1}$ from worker $i - 1$, which is needed for attention computation in the next round.

- Send $O_{j+1}$ and $L_{j+1}$ computed in the previous round to worker $i + 1$.

- Send the already present $Q_j$ to worker $i + 1$.

After receiving $O_j$ and $L_j$, and computing $\Delta O$ and $\Delta L$, we can perform rescaling to update $O_j$, and $L_j$. We describe our distributed attention procedure in Algorithm 1. As demonstrated in Section 4.4, the substantial reduction in communication volume enables complete overlap with computation, effectively eliminating any communication overhead.

**Runtime Analysis** Let $f(\text{query size}, \text{key-value size})$ represent the time required to perform the forward-pass attention computation, and let $comm(\text{tensor size})$ denote the time to transmit a tensor. In LV-XAttn, $Q_i, O_i \in \mathbb{R}^{\frac{S_Q}{n} \times d}$ and $L_i \in \mathbb{R}^{\frac{S_Q}{n}}$ are transmitted during each round. This results in a per-round runtime of:

$$\max\left(f(\frac{S_Q d}{n}, \frac{S_{KV} d}{n}), comm(2 \cdot \frac{S_Q d}{n} + \frac{S_Q}{n})\right) \quad (1)$$

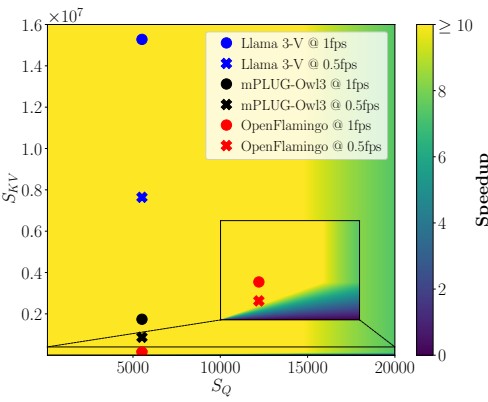

*Figure 4.* The theoretical speedup of LV-XAttn over Ring Attention for cross-attention on a 4-node cluster. Each node is equipped with 4 A100 GPUs, and nodes are interconnected by a 25 GB/s network. The markers represent processing a 2,386-second video and a 3,128-word text prompt – average values for long videos in Video-MME (Fu et al., 2024) – using LLama-3V, mPLUG-Owl3 and OpenFlamingo models at different frame rates. Note that for each frame, a special token `<image>` have to be added to the text-prompt, resulting in $S_Q = 2386 + 3128 = 5514$.

*Table 1.* Runtime analysis for Ring Attention and LV-XAttn. $h$ represents the number of heads in multi-head attention. The attention FLOPs are calculated similar to previous work (Dao, 2024).

|  | Forward | Backward |
|---|---|---|
| Ring Attention | $\dfrac{2 \cdot \frac{S_{KV}}{n} \cdot h \cdot d}{\text{Net Bandwidth}}$ | $\dfrac{4 \cdot \frac{S_{KV}}{n} \cdot h \cdot d}{\text{Net Bandwidth}}$ |
| LV-XAttn | $\dfrac{4 \cdot \frac{S_Q}{n} \cdot \frac{S_{KV}}{n} \cdot h \cdot d}{\text{GPU FLOPS}}$ | $\dfrac{10 \cdot \frac{S_Q}{n} \cdot \frac{S_{KV}}{n} \cdot h \cdot d}{\text{GPU FLOPS}}$ |
| Speedup | $\dfrac{1}{2 \cdot \frac{S_Q}{n}} \dfrac{\text{GPU FLOPS}}{\text{Net Bandwidth}}$ | $\dfrac{2}{5 \cdot \frac{S_Q}{n}} \cdot \dfrac{\text{GPU FLOPS}}{\text{Net Bandwidth}}$ |

In contrast, Ring Attention transmits key-value blocks $K_i, V_i \in \mathbb{R}^{\frac{S_{KV}}{n} \times d}$ in each round, leading to a per-round runtime of:

$$\max\left( f(\frac{S_Q d}{n}, \frac{S_{KV} d}{n}), comm(2 \cdot \frac{S_{KV} d}{n}) \right) \quad (2)$$

Figure 4 shows the theoretical speedup of LV-XAttn over Ring Attention across different $S_Q$ and $S_{KV}$. For MLLM with large visual inputs, where $S_{KV} \gg S_Q$, and slow cross-node interconnect – an unavoidable constraint for inputs exceeding single-node memory capacity – LV-XAttn is compute-bound, while Ring Attention is communication-bound. Consequently, the runtime for LV-XAttn in Equation 1 reduces to $f\left(\frac{S_Q d}{n}, \frac{S_{KV} d}{n}\right)$, while the runtime for Ring Attention in Equation 2 becomes $comm\left(2 \cdot \frac{S_{KV} d}{n}\right)$. A similar analysis applies to the backward pass. Table 1 summarizes the runtime and corresponding speedup for multi-head cross-attention. More in-depth discussion is in Appendix A.

*Table 2.* Evaluated models.

| Model | Num. of CA Layers | Num. of LM Blocks |
|---|---|---|
| Llama 3-V-11b | 8 | 40 |
| mPLUG-Owl3-7b | 4 | 28 |
| mPLUG-Owl3-2b | 4 | 28 |
| mPLUG-Owl3-1b | 4 | 24 |
| OpenFlamingo-9b | 8 | 32 |
| OpenFlamingo-3b | 24 | 24 |

### 3.2. Activation Recomputation for MLLM

In standard attention implementation, during forward pass computation, input tensors $Q_i, K_i, V_i$ and output tensors $O_i$ and $L_i$ are saved for backward pass. However, storing large key-value blocks $K_i$ and $V_i$ increases memory usage, thereby limiting the maximum number of visual inputs that can be processed. For instance, in the case of mPLUG-Owl3-7b with a 3600-frame video, storing $K_i$ and $V_i$ takes 70.08GB per cross-attention layer.

To address this, we observe that while language features $x$ differ across cross-attention layers as they pass through various LM blocks, the visual features $y$ remain unchanged throughout all cross-attention layers, as they are only fed into the cross-attention layers. Thus, instead of storing key-value blocks for each cross-attention layer, we propose to keep a single copy of visual features $y$ that can be accessed by all cross-attention layers. During the backward pass, $y$ is projected to recompute key-value blocks $K_i$ and $V_i$. With $Q_i$ also being recomputed, we only need to save $x, O_i$, and $L_i$ during each cross-attention forward pass.

As demonstrated in the ablation study in Section 4.4, this approach incurs a runtime overhead of less than 8% while enabling the system to handle $1.6\times$ more visual inputs.

## 4. Evaluation

### 4.1. Experimental Setup

**Implementation** LV-XAttn is implemented using PyTorch and Triton (Tillet et al., 2019). It uses `torch.distributed` for distributed communication, while the modified FlashAttention kernels to support rescaling operations are implemented with Triton. We validated the correctness of LV-XAttn by confirming that its output matches that of PyTorch's scaled dot-product attention implementation.

**Model Setup** We evaluate our methods and baselines on 6 models shown in Table 2. Using their publicly available model checkpoints, we replace the cross-attention operations with LV-XAttn and other distributed attention baselines. Following their default configuration, each frame is encoded into 6404 visual tokens for Llama 3-V, 729 vi-

sual tokens for the mPLUG-Owl3 models, and 64 visual tokens for the xOpenFlamingo models. This implies that given the same amount of memory capacity, OpenFlamingo models can accommodate more frames than Llama 3-V and mPLUG-Owl3 models.

For all models, a special token `<image>` must be included in the text prompt for each frame. Consequently, the length of the text prompt must be at least equal to the number of frames in the visual input.

We use a batch size of 1 and fully sharded tensor parallelism for all models to enable a larger context length.

**Cluster Setup** We evaluate our method and baselines on the following configurations: (1) A 16-GPU cluster, each node equipped with 4 A100 80GB GPUs, with the GPUs within a node interconnected via NVLink and a cross-node bandwidth of 25 GB/s, representing a typical setting for cross-node training of up to millions of tokens. (2) An 8-GPU cluster, each node equipped with 1 A30 24GB GPU, with a cross-node bandwidth of 1.25 GB/s, representing a more resource-constrained setup with slower interconnect bandwidth. (3) A 12-GPU cluster, each node equipped with 3 A100 40GB GPUs, with the GPUs interconnected via 64 GB/s PCIe and a cross-node bandwidth of 25 GB/s, used for smaller-scale case studies and ablation studies.

**Baselines** For our method, we use LV-XAttn for the cross-attention layers and Ring Attention for the LM blocks. Our primary baseline is the setup where Ring Attention is used for both the cross-attention layers and LM blocks. We apply our activation recomputation technique to both of these settings for enabling longer context length. We also compare against Deepspeed-Ulysses (Jacobs et al., 2024), which employs sequence parallelism for non-attention layers and head parallelism for attention layers. All methods use FlashAttention.

Our evaluation focuses on comparing the runtime of distributed cross-attention mechanisms. We follow the same benchmarking methodology used in previous work (Dao, 2024; Li et al., 2024), measuring runtime on randomly generated inputs of specific sizes to compare our method with the baselines across different scales. The reported runtime is averaged over 5 trials, following 2 warmup runs.

## 4.2. Comparison with Ring Attention

Table 3 shows the per iteration time of six models using LV-XAttn and Ring Attention on 16 A100 80GB GPUs. For Llama 3-V, LV-XAttn speeds up the cross-attention operation by $17.28 - 27.59\times$. Since the cross-attention operation accounts for the majority of the total iteration time when using Ring Attention, this reduction results in a significant total iteration speedup of $7.51 - 10.62\times$. For the mPLUG-Owl3 and OpenFlamingo models, which process

a larger number of frames (as each frame is encoded into fewer visual tokens compared to Llama 3-V) and thus have longer text lengths (due to the inclusion of a special token `<image>` per frame) and larger $S_Q$, the speedup of cross-attention remains significant, though less pronounced: $7.04 - 15.32\times$ for the mPLUG-Owl3 models and $1.47 - 2.14\times$ for the OpenFlamingo models. Notably, OpenFlamingo-3b, with denser cross-attention layers, spends a larger portion of its time in cross-attention compared to OpenFlamingo-9b when using Ring Attention. Consequently, the speedup in cross-attention translates to a more substantial end-to-end speedup for OpenFlamingo-3b.

Table 4 shows the same experiment on 8 A30 24GB GPUs. In this setup, we have smaller text lengths and fewer frames due to the smaller memory capacity. The speedup for cross-attention operation is greater than that on 16 A100 GPUs: $22.43 - 33.77\times$ for Llama 3-V, $20.6 - 45.85\times$ for the mPLUG-Owl3 models, and $3.97 - 7.2\times$ for the OpenFlamingo models. This is due to smaller query block sizes $\frac{S_Q}{n}$ (shorter computations favors computation-bound LV-XAttn) and slower interconnect bandwidth (longer communication hurts communication-bound Ring Attention), as shown in Table 1. However, the larger cross-attention speedups do not translate into a larger total speedup, as the portion of time spent on cross-attention layers decreases due to slower self-attention layers in LM blocks (caused by lower GPU FLOPS and slower interconnect). Despite this, the total speedup remains $2.2 - 3.13\times$ for Llama 3-V, $1.65 - 3.45\times$ for the mPLUG-Owl3 models, and $1.04 - 2.22\times$ for the OpenFlamingo models.

## 4.3. Comparison with DeepSpeed-Ulysses

For Deepspeed-Ulysses, each attention operation involves two all-to-all communications: one before the computation to gather input query, key and value blocks, and another afterward to distribute attention output along the sequence dimension. The first all-to-all is expensive as it involves communicating the large key-value blocks. To see this, we compare Deepspeed-Ulysses with LV-XAttn on mPLUG-Owl3-2b using the cluster with A100 80GB GPUs. As shown in Table 5, LV-XAttn achieves $1.34 - 1.55\times$ speedup compared to Deepspeed-Ulysses.

In addition, without activation recomputation, the larger memory footprint of Deepspeed-Ulysses limits its ability to process large visual inputs. When running OpenFlamingo-3b on the cluster with A30 24GB GPUs, Table 6 shows that LV-XAttn is able to process more than $4\times$ longer text and visual inputs compared to Deepspeed-Ulysses.

Notably, the head parallelism in Deepspeed-Ulysses restricts both its scalability and flexibility: the maximum degree of parallelism is limited by the number of heads, and the number of heads has to be divisible by the number of workers.

*Table 3.* Per iteration wall-clock time (in seconds) on 16 A100 80GB GPUs with Ring Attention and LV-XAttn. "CA" represents the time spent on cross-attention operations. As $S_Q$ doubles, the cross-attention speedup nearly halves because the runtime for Ring Attention, which is communication-bound, remains constant, while the runtime for LV-XAttn, which is computation-bound, doubles. On the other hand, as $S_{KV}$ doubles, both communication and computation also double, so the speedup remains roughly the same.

| Model | Text length | Frame count | $S_Q$ | $S_{KV}$ | Ring Attention | | LV-XAttn | | Speedup | |
|---|---|---|---|---|---|---|---|---|---|---|
| | | | | | CA (s) | Total (s) | CA (s) | Total (s) | CA | Total |
| Llama 3-V-11b | 2K | 384 | 2K | 2401K | 301.81 | 333.29 | 14.22 | 31.38 | 21.22× | **10.62×** |
| | 2K | 192 | 2K | 1200K | 144.5 | 167.2 | 8.36 | 22.27 | 17.28× | 7.51× |
| | 1K | 192 | 1K | 1200K | 151.83 | 176.16 | 5.5 | 19.93 | **27.59×** | 8.84× |
| mPLUG-Owl3-7b | 8K | 4K | 8K | 2916K | 174.73 | 202.84 | 24.08 | 42.79 | 7.26× | 4.74× |
| | 8K | 2K | 8K | 1458K | 89.88 | 112.28 | 12.14 | 32.72 | 7.41× | 3.43× |
| | 4K | 2K | 4K | 1458K | 92.48 | 107.01 | 6.45 | 19.5 | **14.33×** | **5.49×** |
| mPLUG-Owl3-2b | 8K | 4K | 8K | 2916K | 83.41 | 90.1 | 10.33 | 17.39 | 8.07× | 5.18× |
| | 8K | 2K | 8K | 1458K | 36.66 | 45.42 | 5.21 | 11.28 | 7.04× | 4.03× |
| | 4K | 2K | 4K | 1458K | 37.78 | 44.8 | 2.79 | 8.25 | **13.52×** | **5.43×** |
| mPLUG-Owl3-1b | 8K | 4K | 8K | 2916K | 47.12 | 55.1 | 5.17 | 12.69 | 9.12× | 4.34× |
| | 8K | 2K | 8K | 1458K | 22.63 | 28.81 | 2.62 | 7.99 | 8.64× | 3.6× |
| | 4K | 2K | 4K | 1458K | 23.26 | 29.24 | 1.52 | 5.24 | **15.32×** | **5.58×** |
| OpenFlamingo-9b | 64K | 64K | 64K | 4096K | 95.13 | 165.17 | 62.4 | 126.71 | 1.52× | 1.3× |
| | 64K | 32K | 64K | 2048K | 47.86 | 101.01 | 31.44 | 86.89 | 1.52× | 1.16× |
| | 32K | 32K | 32K | 2048K | 33.58 | 69.53 | 16.0 | 49.5 | **2.1×** | **1.4×** |
| OpenFlamingo-3b | 64K | 64K | 64K | 4096K | 276.69 | 306.51 | 187.45 | 226.04 | 1.48× | 1.36× |
| | 64K | 32K | 64K | 2048K | 138.05 | 166.36 | 94.1 | 120.09 | 1.47× | 1.39× |
| | 32K | 32K | 32K | 2048K | 102.82 | 118.01 | 47.98 | 61.57 | **2.14×** | **1.92×** |

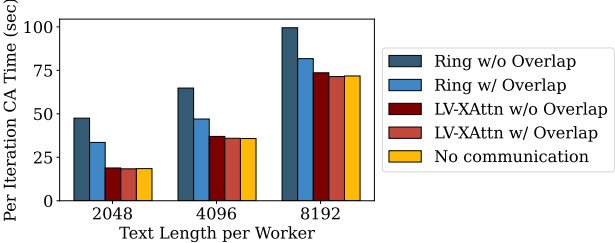

*Figure 5.* Ablation study on the effect of overlapping communication and computation with 6 A100 40GB GPUs. The frame count is set to 2048 per worker. Since processing the same total number of frames on a single GPU is not feasible due to memory constraints, the "no communication" runtime is derived by running the same per-worker input size on a single GPU and then scaling the result by 6. LV-XAttn incurs an overhead of less than 0.42% compared to the no-communication baseline.

## 4.4. Ablation Study

**Overlapping Communication and Computation** Figure 5 shows the time spent on cross-attention in OpenFlamingo-3b using Ring Attention and LV-XAttn, with and without overlapping communication and computation, on 6 A100 40GB GPUs. While overlapping reduces the runtime for Ring Attention, its effect is limited as the large communication overhead of key-value blocks cannot be fully hidden by computation. In contrast, LV-XAttn reduces communication

time by transmitting significantly smaller query, output, and softmax statistics blocks. The overlapping further hides the communication time, enabling distributed attention with no communication overhead.

**Activation Recomputation** Figures 6a and 6b show the iteration time for running mPLUG-Owl-7b and OpenFlamingo-3b on a single node with 3 A100 40GB GPUs, with and without employing activation recomputation for cross-attention layers. By omitting the saving of large key-value blocks, the reduced memory consumption enables the processing of a larger number of frames, increasing by 1.5× and 1.6× for mPLUG-Owl-7b and OpenFlamingo-3b, respectively, with a runtime overhead of less than 8%.

## 5. Discussion

In this section, we discuss the applicability of LV-XAttn to MLLMs with different architectures.

There are two main classes of MLLM architectures: *concatenation-based* and *cross-attention-based*. The first design concatenates tokenized visual inputs with text tokens and feeds them into the LLM backbone. Models such as LLaVA (Liu et al., 2023), InternVL (Chen et al., 2024), NVILA (Liu et al., 2024b), and MiniGPT-4 (Zhu et al., 2024) follow this paradigm. Concatenation-based models have shown strong unified multimodal reasoning capabilities. However, the large number of tokens fed to the

*Table 4.* Per iteration wall-clock time (in seconds) on 8 A30 24GB GPUs with Ring Attention and LV-XAttn. "CA" represents the time spent on cross-attention operations.

| Model | Text length | Frame count | $S_Q$ | $S_{KV}$ | Ring Attention | | LV-XAttn | | Speedup | |
|---|---|---|---|---|---|---|---|---|---|---|
| | | | | | CA | Total | CA | Total | CA | Total |
| Llama 3-V-11b | 512 | 64 | 512 | 400K | 109.07 | 154.18 | 3.35 | 49.24 | 32.56× | **3.13×** |
| | 512 | 32 | 512 | 200K | 55.8 | 101.13 | 2.49 | 45.9 | 22.43× | 2.2× |
| | 256 | 32 | 256 | 200K | 57.5 | 100.75 | 1.7 | 44.93 | **33.77×** | 2.24× |
| mPLUG-Owl3-7b | 1K | 512 | 1K | 364K | 42.41 | 74.28 | 1.42 | 33.32 | 29.96× | **2.23×** |
| | 1K | 256 | 1K | 182K | 20.81 | 50.61 | 0.66 | 30.63 | 31.31× | 1.65× |
| | 512 | 256 | 512 | 182K | 20.84 | 49.89 | 0.45 | 28.95 | **45.85×** | 1.72× |
| mPLUG-Owl3-2b | 1K | 512 | 1K | 364K | 17.85 | 25.94 | 0.78 | 8.71 | 22.89× | **2.98×** |
| | 1K | 256 | 1K | 182K | 9.06 | 16.56 | 0.44 | 7.86 | 20.6× | 2.11× |
| | 512 | 256 | 512 | 182K | 9.15 | 16.34 | 0.3 | 7.44 | **30.39×** | 2.19× |
| mPLUG-Owl3-1b | 1K | 512 | 1K | 364K | 10.6 | 14.41 | 0.44 | 4.18 | 24.25× | **3.45×** |
| | 1K | 256 | 1K | 182K | 5.38 | 8.4 | 0.25 | 3.36 | 21.19× | 2.5× |
| | 512 | 256 | 512 | 182K | 5.31 | 8.22 | 0.18 | 3.03 | **29.44×** | 2.71× |
| OpenFlamingo-9b | 8K | 8K | 8K | 512K | 17.28 | 65.75 | 3.99 | 53.22 | 4.33× | **1.24×** |
| | 8K | 4K | 8K | 256K | 8.74 | 54.09 | 2.2 | 52.17 | 3.97× | 1.04× |
| | 4K | 4K | 4K | 256K | 8.87 | 52.04 | 1.23 | 44.18 | **7.2×** | 1.18× |
| OpenFlamingo-3b | 8K | 8K | 8K | 512K | 52.26 | 69.45 | 12.25 | 32.71 | 4.27× | 2.12× |
| | 8K | 4K | 8K | 256K | 26.09 | 41.73 | 6.43 | 22.22 | 4.06× | 1.88× |
| | 4K | 4K | 4K | 256K | 25.84 | 40.59 | 3.62 | 18.28 | **7.14×** | **2.22×** |

*Table 5.* Per iteration wall-lock time (in seconds) of mPLUG-Owl3-2b ran on A100 80GB GPUs. The model uses multi-head attention with 12 heads.

| Cluster Config. | Text / worker | Frame / worker | DS (s) | LV-XAttn (s) |
|---|---|---|---|---|
| 12 GPUs | 512 | 256 | OOM | **13.38** |
| | 512 | 128 | 12.15 | **8.71** |
| | 256 | 128 | 9.32 | **6.1** |
| 6 GPUs | 512 | 256 | 16.36 | **10.58** |
| | 512 | 128 | 10.41 | **7.09** |
| | 256 | 128 | 8.81 | **5.83** |
| 3 GPUs | 512 | 256 | 15.64 | **10.11** |
| | 512 | 128 | 10.61 | **7.8** |
| | 256 | 128 | 9.91 | **7.37** |

*Table 6.* Per iteration wall-lock time (in seconds) of OpenFlamingo-3b ran on A30 24GB GPUs. The model uses multi-head attention with 8 heads.

| Cluster Config. | Text / worker | Frame / worker | DS (s) | LV-XAttn (s) |
|---|---|---|---|---|
| 8 GPUs | 1K | 1K | OOM | **32.71** |
| | 512 | 512 | OOM | **18.28** |
| | 256 | 256 | OOM | **14.46** |
| 4 GPUs | 1K | 1K | OOM | **19.71** |
| | 512 | 512 | OOM | **13.34** |
| | 256 | 256 | 11.65 | **11.29** |
| 2 GPUs | 1K | 1K | OOM | **13.75** |
| | 512 | 512 | 10.19 | **9.24** |
| | 256 | 256 | 8.04 | **7.87** |

LLM significantly slows down both training and inference, limiting their scalability and efficiency (Ye et al., 2024; Grattafiori et al., 2024).

The second design relies on cross-attention, where cross-attention layers are inserted between LLM layers to incorporate visual features into its intermediate representations. Models such as Flamingo (Alayrac et al., 2022), IDEFICS (Laurençon et al., 2024), Otter (Li et al., 2023a), mPLUG-Owl3 (Ye et al., 2024), Llama 3-V (Grattafiori et al., 2024), and NVLM-X (Dai et al., 2024) adopt this strategy. By bypassing the need to process a large number of visual tokens through the LLM backbone, cross-attention-based MLLMs offer superior computational efficiency when handling large visual inputs (Dai et al., 2024). For example, cross-attention-based MLLMs exhibit 10× better prefill latency than similar-sized concatenation-based MLLMs (Qiu et al., 2025).

However, as the size of visual inputs increase, even cross-attention-based MLLMs encounter efficiency bottlenecks due to communication overhead. To address this, we proposed LV-XAttn, which substantially reduces the communication volume to achieve significant speedup.

To achieve both high accuracy and computational efficiency, recent work have explored a hybrid architecture that uses both concatenation and cross-attention (Dai et al., 2024). We

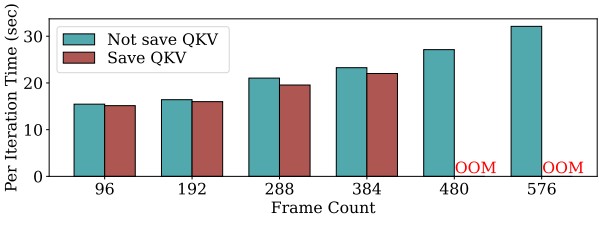

(a) mPLUG-Owl-7b

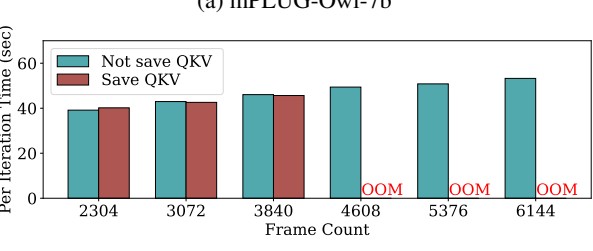

(b) OpenFlamingo-3b

*Figure 6.* Ablation study on the effect of activation recomputation for cross-attention layers with 3 A30 24GB GPUs. Text length is set to $2K$ and $8K$ for mPLUG-Owl-7b and OpenFlamingo-3b, respectively.

note that LV-XAttn can also be applied to such architecture to address communication bottlenecks.

## 6. Related Work

**Memory-efficient Attention** The attention operation has a memory complexity that scales quadratically with sequence length, limiting its scalability for longer contexts. Approximate methods (Kitaev et al., 2020; Zaheer et al., 2020; Beltagy et al., 2020; Choromanski et al., 2021; Ding et al., 2023) and compression techniques (Chevalier et al., 2023; Munkhdalai et al., 2024) reduce memory requirements by sacrificing some model quality. For exact attention, FlashAttention (Dao et al., 2022; Dao, 2024) proposes block-wise computation, which reduces memory complexity to linear while providing runtime speedups by minimizing I/O between GPU HBM and SRAM. However, for applications like long video understanding, which require context lengths exceeding a single GPU's memory capacity, a distributed setup is necessary.

**Parallelism** Distributed attention can be classified into head-parallelism approaches, such as Megatron-LM (Korthikanti et al., 2023a) and Deepspeed-Ulysses (Jacobs et al., 2024), and sequence-parallelism approaches like Ring Attention, its variants DistFlashAttn (Li et al., 2024), and Striped Attention (Brandon et al., 2023). As discussed in Section 2, these methods face significant communication overhead. Our work proposes a solution with minimal communication overhead for cross-attention in MLLMs. General parallelism approaches, such as data parallelism (Dean et al., 2012), pipeline parallelism (Narayanan et al., 2019), tensor

parallelism (Shoeybi et al., 2019), and fully sharded data parallelism (Rajbhandari et al., 2020), can be combined with our approach.

## 7. Conclusion

We introduced LV-XAttn, a distributed, exact cross-attention mechanism for MLLMs with minimal communication overhead. By storing large key-value blocks locally on each worker and transmitting only smaller query blocks, LV-XAttn significantly reduces communication volume, which can be fully hidden by computation. Additionally, the activation recomputation technique reduces memory usage, enabling the processing of longer visual inputs with minimal overhead. Our evaluation demonstrates that LV-XAttn speeds up MLLM iteration by up to $10.62\times$ and enables the processing of visual inputs up to $1.6\times$ longer.

## Acknowledgements

The authors were supported by NSF award CNS 2237306. We thank Zhao Zhang from Rutgers University for providing us access to computing resources. The authors acknowledge the Texas Advanced Computing Center (TACC) at The University of Texas at Austin for providing computational resources that have contributed to the research results reported within this paper. This research used resources of the National Energy Research Scientific Computing Center (NERSC), a Department of Energy Office of Science User Facility using NERSC award DDR-ERCAP0029980. This research also used computational resources from the NSF Cloudlab (Duplyakin et al., 2019) facility. We also thank Wenxuan Tan for his insightful suggestions on improving the paper.

## Impact Statement

This paper aims to improve the efficiency of MLLMs by alleviating communication bottlenecks when processing long visual inputs. By reducing end-to-end runtime, it helps lower the computational cost of training and deploying MLLMs, which in turn can reduce energy consumption and environmental impact. The approach does not alter model outputs, and thus does not introduce additional risks beyond those already associated with MLLMs.

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

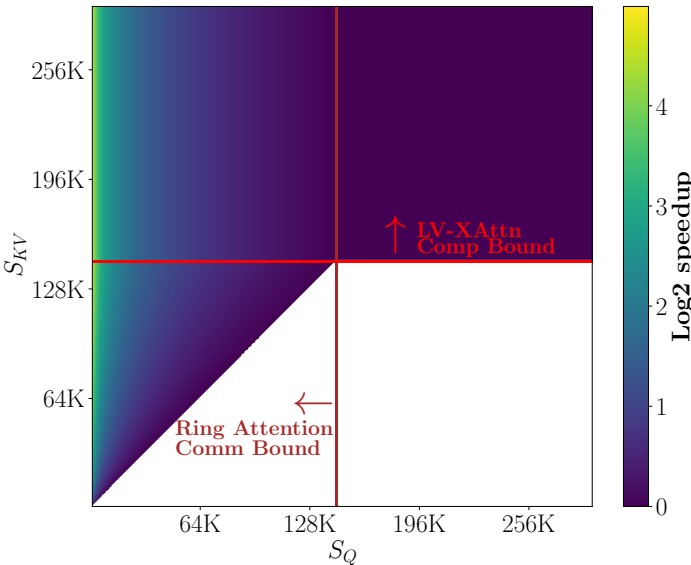

*Figure 7.* The theoretical speedup of LV-XAttn over Ring Attention for on a cluster with 4 nodes, each equipped with 4 A100 GPUs. The uncolored region indicates a speedup of less than 1, meaning LV-XAttn performs slower than Ring Attention. Top-left and bottom-left quadrant represents the typical use case of MLLM with large visual inputs: $S_{KV} \gg S_Q$.

## A. Comparison of LV-XAttn and Ring Attention for General Use Case

We have shown that for applications with large $S_{KV}$ and small $S_Q$ such as long video understanding, LV-XAttn achieves significant speedup over Ring Attention. Here, we provide a more in-depth analysis that generalizes to a broader range of cases.

Figure 7 plots the theoretical speedup of LV-XAttn over Ring Attention for general $S_Q$ and $S_{KV}$. When $S_{KV}$ is large enough (above the horizontal bright red line), the transmission of $Q_i$, $O_i$ and $L_i$ in LV-XAttn is hidden by computation, making LV-XAttn compute-bound. On the other hand, when $S_Q$ is small (to the left of the vertical dark red line), the transmission of $K_i$ and $V_i$ are too large to be hidden by computation, making Ring Attention compute-bound. Their intersection (the top-left quadrant) represents the typical MLLM use case with large visual inputs and small text prompt.

For smaller visual inputs, the reduced $S_{KV}$ causes LV-XAttn to also become communication-bound (the bottom-left quadrant). When both LV-XAttn and Ring Attention are communication-bound, their relative speed depends on communication volume: LV-XAttn sends $2 \cdot \frac{S_Q d}{n} + \frac{S_Q}{n}$, while Ring Attention sends $2 \cdot \frac{S_{KV} d}{n}$. Roughly, when $S_{KV} > S_Q$, LV-XAttn still remains faster than Ring Attention. For MLLMs, each image is encoded into a large number of visual tokens – e.g., 6404 for Llama 3-V, 729 for mPLUG-Owl3 and 64 for OpenFlamingo – so this condition is typically satisfied, making LV-XAttn faster for MLLMs in general.

This also suggests that for self-attention, where $S_Q = S_{KV}$, Ring Attention is preferable. This is why in our experiments, we apply Ring Attention to LM blocks for all baselines. However, when the context length is very large (top-right quadrant), both LV-XAttn and Ring Attention become compute-bound, resulting in identical iteration times, making the choice between them effectively irrelevant.

