# OpenReview forum: "LV-XAttn: Distributed Cross-Attention for Long Visual Inputs in Multimodal Large Language Models"
_ICML.cc/2025/Conference — ICML 2025 poster_

### Official Review · Reviewer_c1fA · 2025-03-11

**Overall Recommendation:** 3

**Summary:**

1. The paper presents LV-XAttn, a distributed cross-attention mechanism designed to address the high memory demands and communication overheads in multimodal large language models (MLLMs) when processing large visual inputs.

2. LV-XAttn reduces communication volume by keeping large key-value blocks locally on each GPU and exchanging smaller query blocks across GPUs, while also introducing an efficient activation recomputation technique to support longer visual context.

3. Evaluations demonstrate that LV-XAttn achieves significant speedups, up to 5.58× end-to-end speedup compared to existing approaches like Ring Attention, making it a more efficient solution for distributed training and inference of MLLMs.

**Claims And Evidence:**

1. Theoretical Analysis: They present a detailed theoretical analysis of the communication benefits of LV-XAttn compared to existing methods like Ring Attention. This includes mathematical formulations of communication volumes and runtime analysis for different scenarios.

2. Empirical Evaluations: The paper includes comprehensive empirical evaluations across multiple models (mPLUG-Owl3 and OpenFlamingo) and cluster configurations (16 A100 GPUs, 8 A30 GPUs, etc.). These evaluations demonstrate significant speedups for both cross-attention operations and overall model iteration time.

3. Ablation Studies: The authors conduct ablation studies to isolate the effects of specific techniques, such as the activation recomputation method. These studies show that the proposed methods achieve their claimed benefits with minimal overhead.

4. Comparison with Baselines: The paper compares LV-XAttn with existing distributed attention mechanisms like Ring Attention and DeepSpeed-Ulysses, demonstrating superior performance in terms of speed and memory efficiency.

**Essential References Not Discussed:**

No

**Experimental Designs Or Analyses:**

1. Main Evaluations Comparing LV-XAttn with Ring Attention
1.1 Design: The authors evaluated LV-XAttn on multiple models (mPLUG-Owl3 and OpenFlamingo) across different cluster configurations (16 A100 GPUs, 8 A30 GPUs, etc.). They measured both cross-attention operation speedup and overall model iteration time speedup.

1.2 Validity: This design is appropriate for demonstrating the practical benefits of LV-XAttn in real-world distributed settings. The use of multiple models and cluster configurations helps establish generalizability.

1.3 Issues: The paper doesn't provide statistical significance testing for the results, which would strengthen the claims. Additionally, while the results are impressive, more detailed analysis of how different parameters (like batch size or sequence length) affect performance could be beneficial.

2. Ablation Studies
2.1 Design: The authors conducted ablation studies to examine the effects of overlapping computation/communication and activation recomputation.
2.2 Validity: These studies effectively isolate specific components of the proposed method and demonstrate their individual contributions. The results (Figures 5 and 6) clearly show the benefits of these techniques.
2.3 Issues: The ablation studies are somewhat limited in scope and could benefit from more detailed analysis of how different parameters affect performance. For example, varying the number of workers or the size of visual inputs could provide additional insights.

**Methods And Evaluation Criteria:**

1. The core idea of LV-XAttn—keeping large key-value blocks locally while exchanging smaller query blocks across GPUs—makes sense given the observation that query blocks are typically much smaller than key-value blocks in MLLMs. This approach directly addresses the communication bottleneck identified in existing distributed attention mechanisms.

2. The activation recomputation technique specifically designed for MLLMs is logical, as it leverages the fact that visual features remain unchanged across cross-attention layers, allowing for memory savings without significant recomputation overhead.

3. The theoretical analysis of communication benefits provides a clear rationale for why LV-XAttn should outperform existing methods like Ring Attention, especially in scenarios with large visual inputs.

4. The use of mPLUG-Owl3 and OpenFlamingo models as test cases is appropriate, as these are representative MLLMs that incorporate cross-attention mechanisms and are known to have challenges with large visual inputs.

5. The evaluation across different cluster configurations (16 A100 GPUs, 8 A30 GPUs, etc.) helps demonstrate the method's effectiveness in various distributed settings and resource constraints.

6.The focus on both cross-attention operation speedup and overall model iteration time speedup provides a comprehensive view of the practical benefits, showing how improvements in one component translate to overall system performance.

**Other Comments Or Suggestions:**

1. Limited Model Scope: The evaluations primarily focus on mPLUG-Owl3 and OpenFlamingo models. While these are representative MLLMs, the approach could benefit from validation on a wider range of multimodal models to demonstrate broader applicability.

2. Lack of Statistical Significance Testing: While the results show substantial speedups, statistical significance testing isn't provided. This could strengthen the claims by quantifying the reliability of the observed performance improvements.

3. Limited Discussion of Trade-offs: The paper could benefit from a more detailed discussion of potential trade-offs, such as increased computational complexity from activation recomputation or limitations in scenarios with very small visual inputs.

4. No Comparison with Other Distributed Cross-Attention Methods: The paper primarily compares with Ring Attention and DeepSpeed-Ulysses but doesn't evaluate against other potential distributed cross-attention mechanisms that might exist in the literature.

**Other Strengths And Weaknesses:**

1. Limited Model Scope: The evaluations primarily focus on mPLUG-Owl3 and OpenFlamingo models. While these are representative MLLMs, the approach could benefit from validation on a wider range of multimodal models to demonstrate broader applicability.

2. Lack of Statistical Significance Testing: While the results show substantial speedups, statistical significance testing isn't provided. This could strengthen the claims by quantifying the reliability of the observed performance improvements.

3. Limited Discussion of Trade-offs: The paper could benefit from a more detailed discussion of potential trade-offs, such as increased computational complexity from activation recomputation or limitations in scenarios with very small visual inputs.

4. No Comparison with Other Distributed Cross-Attention Methods: The paper primarily compares with Ring Attention and DeepSpeed-Ulysses but doesn't evaluate against other potential distributed cross-attention mechanisms that might exist in the literature.

**Questions For Authors:**

1. Limited Model Scope: The evaluations primarily focus on mPLUG-Owl3 and OpenFlamingo models. While these are representative MLLMs, the approach could benefit from validation on a wider range of multimodal models to demonstrate broader applicability.

2. Lack of Statistical Significance Testing: While the results show substantial speedups, statistical significance testing isn't provided. This could strengthen the claims by quantifying the reliability of the observed performance improvements.

3. Limited Discussion of Trade-offs: The paper could benefit from a more detailed discussion of potential trade-offs, such as increased computational complexity from activation recomputation or limitations in scenarios with very small visual inputs.

4. No Comparison with Other Distributed Cross-Attention Methods: The paper primarily compares with Ring Attention and DeepSpeed-Ulysses but doesn't evaluate against other potential distributed cross-attention mechanisms that might exist in the literature.

**Relation To Broader Scientific Literature:**

1. Cross-Attention in Multimodal Models: Cross-attention has been widely adopted in multimodal large language models (MLLMs) for integrating visual information into language backbones. The proposed LV-XAttn mechanism advances this area by specifically addressing the challenges of distributed computation for cross-attention with large visual inputs, which is a critical bottleneck for efficient training and inference.

2. Distributed Attention Mechanisms: Existing distributed attention approaches, such as head-parallelism methods (e.g., Deepspeed-Ulysses) and sequence-parallelism methods (e.g., Ring Attention), face significant communication overheads. LV-XAttn introduces a novel distributed cross-attention mechanism that minimizes communication overhead by keeping large key-value blocks locally on each GPU and exchanging smaller query blocks. This approach is a significant improvement over previous methods, especially for applications with large visual inputs where the size of the query block is much smaller than that of the key-value blocks.

3. Memory-Efficient Attention: The paper's introduction of an efficient activation recomputation technique is related to broader efforts in developing memory-efficient attention mechanisms. Techniques like Flash Attention aim to reduce the memory footprint of attention operations. The activation recomputation method in LV-XAttn specifically targets the memory pressures introduced by storing large key and value tensors for every cross-attention layer in MLLMs, allowing for the processing of longer visual contexts with minimal overhead.

**Theoretical Claims:**

The paper presents several theoretical claims about the communication benefits and speedups of LV-XAttn compared to existing methods like Ring Attention. These claims are supported by mathematical formulations of communication volumes and runtime analysis. The theoretical speedup analysis in Figure 4 and the accompanying equations seem reasonable and align with the intuition that reducing communication volume would lead to significant speedups in distributed settings, especially when dealing with large key-value blocks.

---

> ### Author Rebuttal · Authors · 2025-03-30
>
> Thank you for your time and feedback.
>
> **Q1: Limited model scope.**
>
> A1: Please refer to Q2 of Reviewer zbCn.
>
> **Q2: Lack of statistical significance testing.**
>
> A2: All runtime results are averaged over five iterations, excluding the first two warm-up iterations.
>
> **Q3: Limited discussion of trade-offs.**
>
> A3: In Sections 3.2 and 4.4, we discussed and demonstrated how LV-XAttn’s activation recomputation trades off minimal runtime for reduced memory pressure. For smaller visual inputs, as noted in the Appendix (and discussed in Reviewer EkKR Q3), we expect LV-XAttn to perform similarly to Ring Attention.
>
>
> **Q4: No comparison with other distributed cross-attention methods.**
>
> A4: To the best of our knowledge, existing distributed attention mechanisms either use sequence-parallelism or head-parallelism. Ring Attention and DeepSpeed-Ulysses are two widely adopted and representative mechanisms for each approach, respectively. We believe comparing LV-XAttn to these two methods provides a comprehensive evaluation, as they cover the primary paradigms in distributed cross-attention.

---

### Official Review · Reviewer_c9FC · 2025-03-15

**Overall Recommendation:** 2

**Summary:**

Cross-attention layers takes a significant memory for applications involving large visual inputs, such as long video understanding, making scaling difficult due to high memory requirements. To address this, the authors present LV-XAttn, a distributed and exact cross-attention mechanism that leverages sequence-parallelism with minimal communication overhead. LV-XAttn partitions large key and value blocks across workers while transmitting only small query blocks, enabling blockwise attention computation. This approach significantly reduces communication volume compared to Ring Attention, improving scalability for large-scale visual tasks.

## update after rebuttal
Thank you to the authors for their responses. While some concerns have been addressed, the contributions appear limited to cross-attention operations, making the scenario less compelling. Therefore, I will maintain my current score.

**Claims And Evidence:**

The claims sound reasonable to me. However, I have some concerns and questions about the experiments, as described below.

**Essential References Not Discussed:**

I believe that the paper covers most of the essential references.

**Experimental Designs Or Analyses:**

**Weaknesses and Questions:**
- Why is the proposed method implemented with Ring Attention for the LM blocks? Is LV-XAttn not very efficient for that? Section A in the Appendix provides further analysis (Lines 632–634), but a more detailed investigation is needed to clarify the reasons.
- In the result tables, "CA" indicates that the proposed model achieves a significant speedup in cross-attention operations. Nevertheless, the total speedup is much smaller than that of CA. Where does the proposed model spend more time?

**Methods And Evaluation Criteria:**

**Strengths:**
- The authors evaluated LV-XAttn on various MLLM models and various Cluster Setup.

**Weaknesses and Questions:**
- Which datasets were used for training and evaluation?
- Although the authors demonstrate a significant speedup, there is no discussion on how LV-XAttn affects performance (e.g. accuracy).

**Other Comments Or Suggestions:**

The manuscript requires proofreading. For instance, Figure 2 needs additional clarification on the legends, as they are written entirely in abbreviations, such as ``Fwd Non-CA``.

**Other Strengths And Weaknesses:**

Please check my comments above.

**Questions For Authors:**

Please answer my questions above.

**Relation To Broader Scientific Literature:**

The experimental results may be useful for running models in the specific scenario described in the paper.

**Theoretical Claims:**

The theoretical claims written in Section 3 sound reasonable to me.

---

> ### Author Rebuttal · Authors · 2025-03-30
>
> Thank you for your detailed questions and useful feedback!
>
> **Q1: Which datasets were used for training and evaluation?**
>
> A1: For our experiments, we do not pretrain or finetune the models. Instead, we use the checkpoints from pretrained models and replace their cross-attention operations with LV-XAttn and other distributed attention baselines.
>
> Since LV-XAttn does not affect accuracy, our evaluation focuses on its runtime. We follow the same benchmarking methodology used in Flash Attention and DistFlashAttn to measure the runtime with randomly generated inputs of specific sizes, validating its effectiveness across different scales. It is important to note that our performance benefits are independent of the datasets used.
>
> **Q2: How does LV-XAttn affect the model’s performance (e.g. accuracy)?**
>
> A2: LV-XAttn is an exact distributed attention mechanism and does not change the output of the cross-attention operator. Hence LV-XAttn does not affect the accuracy. In addition, we have validated the correctness of LV-XAttn by confirming that its output matches that of the PyTorch attention implementation. These tests are the same ones used by Flash Attention and DistFlashAttn.
>
> **Q3: Why is Ring Attention used for the LM blocks?**
>
> A3: For self-attention in LM blocks, the short text length and the equal size of query blocks and key-value blocks result in similar performance for LV-XAttn and Ring Attention, making the choice between them less impactful. To confirm this, we conducted additional experiments with mPLUG-Owl3-1b, maintaining a fixed frame count and varying the text length (since LM blocks are only affected by text length). The experiments were performed on a 6-GPU cluster, with each node equipped with three A100 40GB GPUs. The GPUs were interconnected via a 64GB/s PCIe connection, with a cross-node bandwidth of 25GB/s. The results are presented in the table below.
>
> | Text Length | Frame Count | $S_Q$   | $S_{KV}$   | Ring LM + Ring CA (s) | LV-XAttn LM + Ring CA (s) | Ring LM + LV-XAttn CA (s) | LV-XAttn LM + LV-XAttn CA (s) |
> |-------------|-------------|-------|----------|-------------------|-----------------------|-----------------------|---------------------------|
> | 24K        | 1152        | 24K | 820K   | 20.34             | 19.81                 | 16.7                  | 16.06                     |
> | 18K        | 1152        | 18K | 820K   | 17.64             | 16.81                 | 12.91                 | 12.86                     |
> | 12K        | 1152        | 12K | 820K   | 14.37             | 14.35                 | 10.11                 | 9.87                      |
> | 6K        | 1152        | 6K  | 820K   | 13.08             | 12.81                 | 6.68                  | 6.47                      |
>
> Here, “X LM + Y CA” indicates that the LM blocks are implemented with X, and the cross-attention layers are implemented with Y. As shown in the table, there is minimal difference between LM blocks implemented with Ring Attention or LV-XAttn. The speedup observed is primarily due to how the cross-attention layers are implemented.
>
> **Q4: Why is the end-to-end speedup much smaller than the speedup of cross-attention?**
>
> A4: The total iteration time includes the time spent on both cross-attention and non-cross-attention operations within the MLLM. LV-XAttn significantly accelerates the cross-attention layers, but the time spent on the remaining components of the MLLM remains unchanged. This is illustrated in Figure 2, where LV-XAttn reduces both the forward and backward cross-attention times (FWD CA and BWD CA), while the forward and backward times for vision and non-cross-attention layers (FWD Vision, FWD Non-CA, and BWD Non-CA) stay the same. As a result, the overall end-to-end speedup is limited by the time spent on non-cross-attention layers, meaning the speedup is smaller than that observed in the cross-attention layers. This is particularly true for mPLUG-Owl3 models, which only have 4 cross-attention layers out of a total of 24 or 28 LM blocks.
>
> **Q5: Figure 2 needs additional clarification on the legends.**
>
> A5: The abbreviations used in the legend for Figure 2 are as follows:
> * FWD CA: Forward pass through the cross-attention layers.
> * BWD CA: Backward pass through the cross-attention layers.
> * FWD Vision: Forward pass through the vision encoder.
> * FWD Non-CA: Forward pass through the non-cross-attention layers.
> * BWD Non-CA: Backward pass through the non-Cross-attention layers.
>
> We will ensure these clarifications are included in our revision.

---

### Official Review · Reviewer_EkKR · 2025-03-17

**Overall Recommendation:** 3

**Summary:**

The paper introduces ​LV-XAttn, a distributed cross-attention mechanism designed to handle large visual inputs in ​Multimodal Large Language Models (MLLMs) with minimal communication overhead. Cross-attention is commonly used in MLLMs to integrate visual information into the language backbone, but processing large visual inputs (e.g., long videos) leads to high memory demands and significant communication costs in distributed setups. Existing distributed attention mechanisms, such as ​head-parallelism and ​sequence-parallelism, face scalability and efficiency challenges, particularly with large key-value blocks. The key contributions of the paper:
1. The proposed mechanism keeps large key-value blocks locally on each GPU and exchanges smaller query blocks across GPUs, significantly reducing communication volume. This approach leverages the observation that in applications with large visual inputs, the query block size is typically much smaller than the key-value blocks.
​2. To further reduce memory pressure, LV-XAttn introduces a technique where visual tokens are shared across cross-attention layers, and activations are recomputed during the backward pass. This allows processing longer visual inputs with minimal overhead (less than 8%).

**Claims And Evidence:**

The claims made in the submission are generally supported by clear and convincing evidence, as the paper provides theoretical analysis, empirical evaluations, and comparisons with existing methods.

**Essential References Not Discussed:**

None

**Experimental Designs Or Analyses:**

The paper provides a detailed algorithm (Algorithm 1) for the forward pass of LV-XAttn, which is helpful for reproducibility. However, it would be beneficial to include more details on the implementation, such as the specific libraries and frameworks used, to facilitate replication of the results.

**Methods And Evaluation Criteria:**

The proposed methods and evaluation criteria are well-designed and appropriate for the problem at hand. LV-XAttn effectively addresses the bottlenecks of cross-attention in MLLMs, and the evaluation demonstrates its advantages in terms of efficiency, scalability, and practicality for real-world applications.

**Other Comments Or Suggestions:**

None

**Other Strengths And Weaknesses:**

Weakness：
​Limited Applicability to Certain MLLM Architectures: The proposed method is specifically designed for MLLMs that utilize cross-attention mechanisms to process visual tokens. Consequently, it may not be directly applicable to other MLLM architectures, such as ​LLaVA, which employ different approaches for integrating visual information. This limitation restricts the generalizability of LV-XAttn to a broader range of multimodal models.

​Lack of Analysis on Single Image Processing: In real-world deployment scenarios, MLLMs are often required to process not only long videos but also individual images. The paper does not provide an analysis of the runtime performance or potential trade-offs when applying LV-XAttn to single-image inputs. This omission raises concerns about whether the proposed method might introduce inefficiencies or negative effects in such cases, which are critical for practical applications. A more comprehensive evaluation encompassing both long videos and single images would strengthen the paper's relevance to real-world use cases.

**Questions For Authors:**

None

**Relation To Broader Scientific Literature:**

None

**Theoretical Claims:**

The paper presents theoretical claims and analyses, particularly regarding the communication benefits and runtime efficiency of ​LV-XAttn compared to ​Ring Attention. I have checked the runtime analysis and don't find problems, and the empirical results support the claims.

---

> ### Author Rebuttal · Authors · 2025-03-30
>
> Thank you for your insightful questions and feedback!
>
> **Q1: Details on implementation.**
>
> A1: LV-XAttn is implemented using PyTorch and Triton. It uses `torch.distributed` for distributed communication, while the modified FlashAttention kernels to account for rescaling operations are implemented with Triton. We will add these details to the paper. We plan to open-source LV-XAttn.
>
> **Q2: Limited applicability to certain MLLM architectures.**
>
> A2: Please refer to Q2 of Reviewer zbCn.
>
> **Q3: ​Lack of analysis on single image processing.**
>
> A3: LV-XAttn addresses the communication overhead caused by the large number of visual tokens. While this issue is common in long video inputs due to the high frame count, it can also arise in single-image inputs. This can be due to either a large image size (for example, in mPLUG-Owl3, processing a single 3,840×3,840 high-resolution image is equivalent to processing a 100-frame video of standard frame size) or a high number of visual tokens per image (for instance, Llama-3V encodes each image into 6,404 visual tokens, whereas OpenFlamingo uses only 64, making a single image in Llama-3v equivalent to a 100-frame video in OpenFlamingo).
>
> In both video and single-image inputs, these scenarios result in large key-value blocks, which are the primary source of communication overhead in existing distributed attention mechanisms. LV-XAttn effectively addresses this challenge.
>
> For single images with small image size or low visual token count, we note that distributed attention may not be necessary. Even if LV-XAttn is applied in such scenarios, we expect its performance to be comparable to Ring Attention, as discussed in Appendix A.

---

### Official Review · Reviewer_zbCn · 2025-03-18

**Overall Recommendation:** 3

**Summary:**

This paper presents LV-XAttn, a distributed cross-attention mechanism with low communication overhead that can significantly reduce the inference time of MLLMs adopting cross-attn strategy. The authors also introduce the activation recomputation technique enabling supporting longer visual inputs. The proposed LV-XAttn is more efficient compared to Ring-Attention.

**Claims And Evidence:**

The claims are clear. For example in Figure 2, the comparison of the overheads is well illustrated and evidenced.

**Essential References Not Discussed:**

The references are sufficient. However, I noticed that the Llama 3V, a MLLM which is typical and also adopt the cross-attention paradigm to  perceive visual inputs is not discussed in this paper.

**Experimental Designs Or Analyses:**

I have checked the soundness and validity of the experimental designs. For example the Model Setup (i.e., using mPLUG-Owl3, OpenFlamingo), the Cluster Setup (i.e., using A100 80G GPUs), and the Baselines (i.e., Ring Attention, and Deepspeed-Ulysses). There are no issues.

**Methods And Evaluation Criteria:**

The evaluation adopted in this paper is sense-making.

**Other Comments Or Suggestions:**

Suggestions
- The data with statistics characteristics used in the evaluation of Figure 2 should be included in the caption.

**Other Strengths And Weaknesses:**

Strengths
- The strength of this paper is very clear: the proposed cross-attention mechanism can reduce the communication overhead for MLLMs leveraging cross-attention paradigm on video scenario.

Weakness
- I think the proposed method is an extend version of the Ring-Attention on videos scenario, which is specifically effective for MLLMs who adopt the cross-attention paradigm to perceive visual inputs. However, the mainstream architecture adopted in current MLLMs is the concatenation paradigm, which prepends the visual tokens to prompt tokens directly and feeds them to LLM layers. Therefore I think the applicable scenarios are limited for the LV-XAttn.

**Questions For Authors:**

Could the proposed LV-XAtten also improve the efficiency for MLLMs using concatenation paradigm?

**Relation To Broader Scientific Literature:**

The technique proposed in this paper, which speedup the MLLMs on long videos inputs are useful.

**Theoretical Claims:**

Theoretical claims presented in this paper are reasoning. For example the Figure 4 depicting the theoretical speedup of LV-XAttn over Ring Attention.

---

> ### Author Rebuttal · Authors · 2025-03-30
>
> Thank you for the constructive feedback and questions!
>
> **Q1: Llama-3V is not discussed in the paper.**
>
> A1: Thank you for pointing this out! Llama-3V [1] also utilizes a cross-attention architecture, allowing LV-XAttn to be applied to it for large visual inputs. We conducted additional experiments on Llama-3V (`Llama-3.2-11B-Vision-Instruct`), comparing LV-XAttn and Ring Attention, following a similar setup to Section 4.2. The experiments were performed on a 6-GPU cluster, with each node equipped with three A100 40GB GPUs. The GPUs inside a node were interconnected via a 64GB/s PCIe connection, with a cross-node bandwidth of 25GB/s. The results are presented in the table below.
>
> | Model              | Text Length | Frame Count | $S_Q$  | $S_{KV}$   | Ring Attention CA (s) | Ring Attention Total (s) | LV-XAttn CA (s) | LV-XAttn Total (s) | Speedup CA | Speedup Total |
> |--------------------|------------|-------------|-----|-------|-----------------------|-------------------------|-----------------|-------------------|------------|---------------|
> | Llama-3V      | 384         | 120          | 384  | 750K | 23.30                | 37.54                  | 0.43           | 14.45             | 54.19$\times$      | 2.60$\times$        |
> | Llama-3V      | 192         | 120          | 192  | 750K | 23.01                | 37.21                  | 0.40           | 14.40             | 57.53$\times$      | 2.58$\times$        |
> | Llama-3V      | 192         | 60          | 192  | 375K | 11.69               | 22.92                  | 0.22           | 11.61             | 53.14$\times$      | 1.97$\times$        |
>
> For Llama-3V, each frame is encoded into 6,404 visual tokens. In comparison, OpenFlamingo and mPLUG-Owl3 models encode each frame into 64 and 729 visual tokens, respectively. This results in an even greater imbalance between query block size and key-value block size for Llama-3V. Since LV-XAttn transmits the smaller query blocks while Ring Attention transmits the larger key-value blocks, LV-XAttn achieves a significant speedup over Ring Attention.
>
> **Q2: Applicability to MLLMs with alternative architecture.**
>
> A2: Recent MLLMs adopt two main classes of architecture: cross-attention based, where visual information is fused with text information through cross-attention, and concatenation based, where visual tokens are directly concatenated with text tokens and fed into the LLM backbone as inputs. While LV-XAttn is designed to address the communication overhead of the former class of MLLMs, it might still be useful for concatenation based MLLMs for the following reasons.
>
> First, to generate visual tokens, concatenation based MLLMs often use a visual adapter to align visual features generated by the vision encoder with textual prompt. Cross-attention is widely adopted in these visual adapters (e.g. BLIP-2 [2], Qwen-VL [3], Video-Llama [4], InternVL [5]). LV-XAttn can be used in the visual adapters when processing large visual inputs.
>
> Second, with large amounts of visual tokens fed into the LLM backbones, concatenation based MLLMs also require distributing the attention operation. As discussed in Appendix A, with large context size, LV-XAttn can be used for distributed attention without slowdown when compared to Ring Attention.
>
> Finally, we would also like to note that cross-attention-based architectures are widely used in recent models, such as Llama-3V [1] and NVLM-H [6], due to their computational efficiency during training and inference, and due to their ability to prevent degradation for text-only tasks [1, 6]. As a result, addressing the communication bottleneck in these models with LV-XAttn is important for improving their overall efficiency.
>
> **Q3: Figure 2 setup clarification.**
>
> A3: For Figure 2, mPLUG-Owl-7b was run with a text length of 4K and a frame count of 2K ($S_Q=4K$, $S_{KV}=1458K$), while OpenFlamingo-3b was run with a text length and frame count of 32K ($S_Q=32K, $S_{KV}=2048K$). These results come from the same experiment presented in Table 3. We will ensure these clarifications are included in the caption in our revision.
>
> [1] The Llama 3 Herd of Models: https://arxiv.org/abs/2407.21783
>
> [2] BLIP-2: Bootstrapping Language-Image Pre-training with Frozen Image Encoders and Large Language Models: https://proceedings.mlr.press/v202/li23q/li23q.pdf
>
> [3] Qwen-VL: A Versatile Vision-Language Model for Understanding, Localization, Text Reading, and Beyond: https://arxiv.org/pdf/2308.12966
>
> [4] Video-LLaMA: An Instruction-tuned Audio-Visual Language Model for Video Understanding: https://aclanthology.org/2023.emnlp-demo.49/
>
> [5] InternVL: Scaling up Vision Foundation Models and Aligning for Generic Visual-Linguistic Tasks: https://openaccess.thecvf.com/content/CVPR2024/papers/Chen_InternVL_Scaling_up_Vision_Foundation_Models_and_Aligning_for_Generic_CVPR_2024_paper.pdf
>
> [6] NVLM: Open Frontier-Class Multimodal LLM: https://arxiv.org/pdf/2409.11402

---

### Decision · Program_Chairs · 2025-05-01

**Decision:**

Accept (poster)

**Comment:**

This paper presents LV‑XAttn, which exchanges small query blocks while keeping large key/value blocks local and using activation recomputation to extend visual context. The paper presents theoretical rationale and empirical speedups (up to 5.6× over Ring Attention and DeepSpeed‑Ulysses on mPLUG‑Owl3 and OpenFlamingo across diverse GPU clusters). Reviewers highlight that LV‑XAttn’s benefits are confined to cross‑attention MLLM architectures, leaving out concatenation‑based models (e.g. LLaVA, Llama 3V), and that its performance on single‑image inputs, accuracy trade‑offs, and statistical significance are not evaluated. They also call for broader model coverage, more detailed implementation/runtime and accuracy analyses, and minor clarifications in figures, legends, and experimental descriptions. In light of its significant contributions but narrow applicability and the need for additional validation and clarity, I recommend a weak accept. And I strongly encourage the authors to revise the paper by following reviewers' suggestions.